

# The significance of PD-1/PD-L1 imbalance in ulcerative colitis

Wei Shi[1,2,*], Yu Zhang[2,*], Chonghua Hao[2], Xiaofeng Guo[3], Qin Yang[4], Junfang Du[1], Yabin Hou[2], Gaigai Cao[2], Jingru Li[2], Haijiao Wang[5] and Wei Fang[1]

[1] Department of Clinical Laboratory, Shanxi Provincial People's Hospital, Taiyuan, China
[2] The Fifth Clinical Medical College of Shanxi Medical University, Taiyuan, China
[3] Department of Gastroenterology, Shanxi Provincial People's Hospital, Taiyuan, China
[4] Department of Pathology, Shanxi Provincial People's Hospital, Taiyuan, China
[5] Shanxi Center for Disease Control and Prevention, Taiyuan, China
* These authors contributed equally to this work.

Corresponding author
Chonghua Hao,
teacherhaochonghua@163.com

## ABSTRACT

**Objectives:** To investigate the expression and significance of programmed cell death protein 1 (PD-1) and programmed cell death ligand-1 (PD-L1) in the mucosal tissues and peripheral blood of patients with ulcerative colitis (UC).

**Methods:** Eighty patients with UC were recruited from January 2021 to August 2022 from the Shanxi Province People's Hospital. PD-1 and PD-L1 expression was assessed by immunohistochemistry in mucosal tissues. An enzyme-linked immunosorbent assay was used to measure soluble PD-1 and PD-L1 levels in peripheral blood serum, and the membrane-bound forms of PD-1 (mPD-1), (T-helper cell) Th1 and Th17, in peripheral blood were determined by flow cytometry.

**Result:** PD-1 expression was observed only in the monocytes of the mucosal lamina propria of UC patients, while PD-L1 was mainly located in both epithelial cells and monocytes on the cell membrane. The expression level of PD-1/PD-L1 in the monocytes and epithelial cells of mucosal lamina propria increased with disease activity ($P < 0.05$). The percentages of PD-1/T and PD-1/CD4+T in the peripheral blood of moderate UC patients (PD-1/T 12.83 ± 6.15% and PD-1/CD4+T 19.67 ± 9.95%) and severe UC patients (PD-1/T 14.29 ± 5.71% and PD-1/CD4+T 21.63 ± 11.44%) were higher than in mild UC patients (PD-1/T 8.17 ± 2.80% and PD-1/CD4+T 12.44 ± 4.73%; $P < 0.05$). There were no significant differences in PD-1/CD8+T cells between mild and severe UC patients ($P > 0.05$). There was a statistically significant difference in the expression level of sPD-L1 between the UC groups and healthy controls, and the expression level of sPD-L1 increased with disease severity ($P < 0.05$); however, there was no statistically significant difference in sPD-1 expression levels between the UC groups and healthy controls ($P > 0.05$). The correlation coefficients between Th1 and sPD-L1, PD-1/T, PD-1/CD4+T and PD-1/CD8+T were 0.427, 0.589, 0.486, and 0.329, respectively ($P < 0.001$). The correlation coefficients between Th17 and sPD-L1, PD-1/T, PD-1/CD4+T and PD-1/CD8+T were 0.323, 0.452, 0.320, and 0.250, respectively ($P < 0.05$).

**Conclusion:** The expression level of PD-1/PD-L1 was correlated with UC disease activity, and two forms of PD-1 and PD-L1 may be used as a potential marker for predicting UC and assessing disease progression in UC patients. PD-1/PD-L1 imbalance was a significant phenomenon of UC immune dysfunction. Future

research should focus on two forms of PD-1/PD-L1 signaling molecules to better understand the pathogenesis of UC and to identify potential drug therapies.

# INTRODUCTION

Ulcerative colitis (UC) is a chronic relapsing inflammatory bowel disease (IBD) with rectal bleeding, diarrhoea, and abdominal pain as its main symptoms and inflammatory cell infiltration, diffuse crypt abnormalities, and diffuse superficial ulcers as its main histopathological features (*Feakins, 2014*). IBD is a global disease with a high incidence and prevalence throughout the world (*Molodecky et al., 2012*). Genetic susceptibility, environmental factors, gut microbiota dysbiosis and immune dysbiosis contribute to the pathogenesis of UC (*Ananthakrishnan, 2015*). Although the exact causes of UC are unclear, there has been a growing interest in uncontrolled immune responses as an important factor (*Tatiya-Aphiradee, Chatuphonprasert & Jarukamjorn, 2018*). Current treatments for UC include salicylic acid preparations, glucocorticoids, biological agents, fecal bacteria transplantation, and surgical resection, and many UC patients have benefited from these therapies (*Ungaro et al., 2017*). Antitumor necrosis factor antibodies are a novel therapy for UC that have gained recent attention, but antibody therapy is not effective for all patients, and can actually lead to an increased risk of infectious complications in some patients (*Katsanos & Papadakis, 2017*; *Click & Regueiro, 2019*). It is therefore imperative to identify novel therapeutic targets beyond immune suppression for the treatment of UC and IBD.

Programmed cell death-1 (PD-1) and programmed cell death ligand-1 (PD-L1) are members of the CD28 superfamily and the B7 superfamily, respectively. PD-1 emits negative signals when it interacts with PD-L1. Both PD-L1 and PD-1 are expressed most prominently on activated CD4+ and CD8+ T cells, and their interaction inhibits activated CD4+ and CD8+T cell proliferation and mediates immune tolerance or causes a harmful effect on antitumor immunity, contributing to immune evasion (*Pinchuk et al., 2008*; *Wang & Wu, 2020*). Previous research has found that these negative co-stimulators play a critical role in innate and adaptive immune responses and in gut homeostasis (*Chulkina, Beswick & Pinchuk, 2020*). At the same time, recent studies on PD-1/PD-L1 have also made new progress in UC, and PD-1/PD-L1 may be a potential therapeutic target for UC (*Roosenboom et al., 2021*; *Cassol et al., 2020*). PD-1 and PD-L1 also exist in soluble forms. Soluble programmed cell death ligand-1 (sPD-1) is encoded by PD-1Deltaex3, which lacks the transmembrane region and has its own immune regulatory function; like a cytokine, it plays a role in aberrant T-cell proliferation (*Dai et al., 2014*). Soluble programmed cell death ligand-1 (sPD-L1) is mainly produced by the cleavage of PD-L1. SPD-1 and sPD-L1 have important immune regulatory functions which can bind specifically to sPD-1 and PD-1 (*He et al., 2020*). However, the changes of sPD-1/sPD-L1 in UC peripheral blood and

their role in immune dysfunction need to be explored as PD-1/PD-L1 likely plays an important role in cellular immunity dysfunction in UC. In our study, mucosal tissue and peripheral blood samples from UC patients were used to investigate the clinical value of two forms of PD-1/PD-L1 in UC.

## MATERIALS AND METHODS

### Setting and study design

In this prospective cohort study, 80 patients with ulcerative colitis (UC), 30 healthy controls (HC) and 20 patients with acute enteritis were recruited from the Department of Gastroenterology at the Shanxi Province People's Hospital, which a general teaching hospital affiliated with Shanxi Medical University. Shanxi Provincial People's Hospital is a comprehensive tertiary hospital that provides medical services to the people of Shanxi Province, serving about 370,000 people each year. Patients were recruited for this study between February 2021 to August 2022. Patients were included in this study if they met the diagnostic requirements outlined by the Chinese consensus on the diagnosis and treatment of inflammatory bowel disease (2018, Beijing) (*Inflammatory Bowel Disease Group, Chinese Society of Gastroenterology, Chinese Medical Association, 2018*). Patients were excluded if they had a history of Crohn's disease, rheumatoid arthritis, infectious diseases, intestinal tuberculosis, recent surgery, malignancies, and or if they had used immunosuppressive drugs before. This study was approved by the Ethical Committee of Shanxi Provincial People's Hospital, Shanxi Medical University (Ethical Application Ref: 2021-18) and written, informed consent was obtained from all study participants.

### Immunohistochemistry

The immunohistochemistry methods were as follows: the fixed samples were embedded in neutral buffered formalin for 6–12 h. They were then washed, dehydrated, and embedded in paraffin and cut into three-μm-thick sections. Immunoperoxidase staining was then performed with antibodies against PD-1 (Clone MX033; ready to use) and PD-L1 (Clone E1L3N; 1:200) in addition to HE staining. PD-1 expression was found to be restricted to inflammatory cells in the lamina propria, but not on epithelial cells. However, PD-L1 was found on both epithelial and inflammatory cells in the lamina propria. Cells expressing PD-1- and PD-L1-positive markers were expressed as continuous variables, ranging from 0 to 100, and as categorical variables, divided into four categories by staining intensity: 0 (negative): 1% of cells stained, one (weak): 1–5% of cells stained, two (moderate): 5–10% of cells stained, and three (strong): >10% of cells stained. Healthy controls underwent colonoscopy for polyp surveillance. Healthy controls did not present endoscopic abnormalities reflecting inflammation and pathological findings revealed no active inflammatory response in mucosal tissues. Specimens were obtained from patients with acute enteritis who had a pathological diagnosis of acute inflammatory response in intestinal mucosal tissue.

## Monoclonal antibodies and flow cytometry

The following monoclonal antibodies and reagents were used in this study: flurescein isothiocyanate (FITC)-conjugated anti-CD3, allophycocyanin (APC)-CY7-conjugated anti-CD4, allophycocyanin (APC)-conjugated anti-CD8, (PerCP)-CY5.5-conjugated anti-CD45, PE-conjugated mouse IgG isotype control, PE-conjugated anti-programmed cell death-1 (all from Caprico Biotechnologies, China), Anti-Human IFN-APC, and Anti-Human IL-17A-PE. Flow cytometry was used to detect PD-1, Th1, and Th17 in peripheral blood. Blood samples were treated with hemolysin to obtain mononuclear cells. Antibodies were used to label CD3+T cells, CD4+T cells and CD8+T cells, and the percentage of PD-1 on mononuclear cells γ was analyzed by flow cytometry. BD FACSARIAII and BeamCyte were used for the Flowcytometric analysis of stained cell suspensions.

## ELISA

Blood samples were temporarily stored in a yellow test tube before testing, and then centrifuged for 3,000 r/min for 15 min. The serum was taken and stored in a test tube without endotoxin at $-80\,°C$. All reagents and samples were removed from the refrigerator 60 min before measurement and returned to room temperature. The sPD-1 and sPD-L1 reagents were purchased from RuiXin Biotech, and detailed experimental procedures were outlined in the reagent instructions. Interferon $\gamma$ was analyzed using an automatic chemiluminescence analyzer (WanTai Caris-200).

## Statistical analysis

Measurement data conforming to normal distribution were expressed as meanstandard deviation. The T-test or analysis of variance were used to compare data between groups. Non-normal measurement data were expressed as medians with interquartile ranges (IQR). The Mann–Whitney U test was used to compare continuous variables of two independent groups, and the Kruskal–Wallis H test was used to compare multiple independent samples. SPSS version 23.0 (IBM Corp., Armonk, NY, USA) was used for statistical analysis. The data were considered to be statistically significant when $P < 0.05$.

# RESULTS

## Baseline patient characteristics

There were 80 UC patients included in this study: 25 mild UC cases (14 males/11 females), with an age range of 23 to 68 years and a mean age of 46.84 ± 12.84 years; 30 moderate UC cases (17 males/13 females), with an age range from 18 to 66 years and a mean age of 45.93 ± 13.70 years; and 25 severe UC cases (18 males/seven females), with an age range from 24 to 72 years and a mean age of 48.92 ± 13.58 years. The control group included 20 acute enteritis patients (nine males/11 females), with an age range of 28 to 66 years and a mean age of 44.88 ± 10.88 years, and 30 healthy controls (16 males/14 females), with an age range of 21 to 67 years and a mean age of 44.07 ± 12.17 years. The mean age did not differ significantly between the experimental and control groups. It was noteworthy that WBC,

**Table 1 Baseline patients characteristics.** There are 80 UC patients including 25 mild UC cases. The age of mild UC cases ranged from 23 to 68 years and the mean age was 46.84 ± 12.84 years, 14 male cases; 30 moderate UC cases, the age of moderate UC ranged from 18 to 66 years and the mean age was 45.93 ± 13.70 years, 17 male cases; 25 severe UC cases, the age of severe UC ranged from 24 to 72 years and the mean age was 48.92 ± 13.58 years, 18 male cases; 20 acute enteritis patients cases, the age of acute enteritis patients ranged from 28 to 66 years and the mean age was 44.88 ± 10.88 years, nine male cases and 30 healthy control cases, the age of healthy control ranged from 21 to 67 years and the mean age was 44.07 ± 12.17 years, 16 male cases. The mean age did not differ significantly between the experimental and control groups. It was noteworthy that WBC, Albumin and K+ were statistically significant in the UC group, acute enteritis group and healthy control group.

| | HC ($n$ = 30) | Acute enteritis ($n$ = 20) | Mild UC ($n$ = 25) | Moderate UC ($n$ = 30) | Severe UC ($n$ = 25) |
|---|---|---|---|---|---|
| Gender | | | | | |
| Female | 14 | 11 | 11 | 13 | 7 |
| Male | 16 | 9 | 14 | 17 | 18 |
| Age | 44.07 12.17 | 44.88 10.88 | 46.84 12.84 | 45.93 13.70 | 48.92 13.58 |
| BMI | 21.76 3.08 | 20.93 3.11 | 22.68 2.83 | 22.30 2.57 | 21.74 3.30 |
| Smoking status | | | | | |
| Yes | 7 | 4 | 3 | 5 | 8 |
| No | 23 | 16 | 22 | 25 | 17 |
| Drinking | | | | | |
| Yes | 12 | 5 | 4 | 3 | 6 |
| No | 18 | 15 | 21 | 27 | 19 |
| WBC (×10$^9$/L) | 5.72 1.16 | 6.57 1.33 | 5.76 1.75 | 6.48 3.07 | 8.77 3.20 |
| Albumin | 45.36 6.44 | 42.60 7.46 | 41.99 6.78 | 34.54 7.91 | 31.39 5.88 |
| K$^+$ | 4.08 0.27 | 3.54 0.42 | 3.88 0.33 | 3.56 0.26 | 3.48 0.50 |
| Course (month) | None | 0.5 (0.33,0.88) | 39 (24,48) | 36 (24,72) | 36 (14.25,135) |
| Location | | | | | |
| E1: ulcerative proctitis | None | None | 14 | 3 | 2 |
| E2: left-sided UC | None | None | 3 | 8 | 4 |
| E3: extensive UC | None | None | 8 | 19 | 19 |

Albumin and K+ were statistically significant in the UC group, acute enteritis group and healthy control group (Table 1).

## PD-1/PD-L1 was specifically expressed in the mucosal tissues of UC patients, but not in acute enteritis patients or healthy controls

PD-1 was only expressed in monocytes located on the mucosal lamina propria of UC patients, while PD-L1 was discovered in both epithelial cells and in monocyte membranes. Negative PD-1/PD-L1 expression was observed in normal mucosa and in the mucosa of common acute enteritis patients. Programmed cell death-1 (PD-1)/programmed cell death-ligand 1 (PD-L1) expression in the colon mucosa of healthy controls (HC), acute enteritis patients and ulcerative colitis (UC) patients (200×) are shown in Fig. 1. Normal colon mucosa shown in Figs. 1A–1C show PD-1 and PD-L1-negative monocytes without PD-L1 staining in the epithelium. Figure 1D–1F shows the colon mucosa of acute enteritis patients with PD-1/PD-L1-positive monocytes, but no epithelial cells. Colon mucosa of

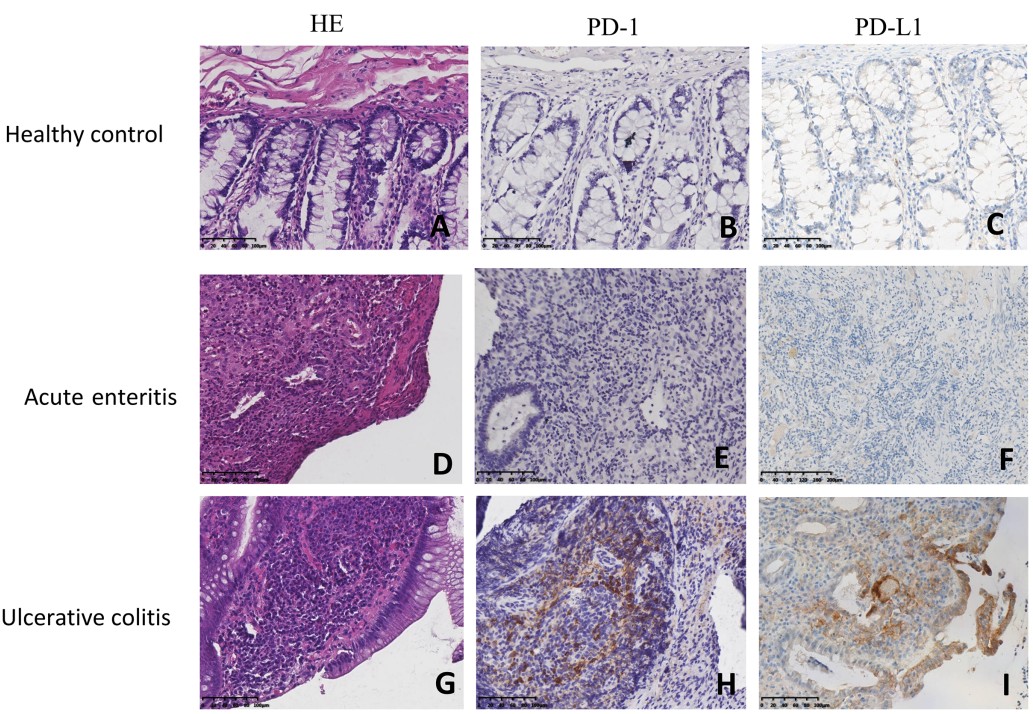

**Figure 1 Localization of PD-1 and PD-L1 in colitis.** Programmed cell death-1 (PD-1) /programmed cell death-ligand 1 (PD-L1) expression in colon mucosa of healthy control (HC), Acute enteritis and ulcerative colitis (UC) (200×). (A–C) Normal colon mucosa cases showing PD-1-negative and PD-L1-negative monocyte, and no PD-L1 staining in the epithelium. (D–F) Acute enteritis with very few PD-1/PD-L1-positive monocyte, but not in the epithelium. (G–I) Ulcerative colitis specimen showing numerous PD-1-positive monocyte and strong PD-L1 expression in the monocyte and epithelium cells.

ulcerative colitis patients had numerous PD-1-positive monocytes and a strong expression of PD-L1 in the monocytes and epithelium, as shown in Figs. 1G–1I.

## The immunohistochemical analysis showed the expression of PD-1/PD-L1 in the mucosal tissues of UC patients was affected by the degree of UC disease inflammation

The expression of PD-1/PD-L1 was negative in healthy controls, while acute enteritis patients had very few PD-L1-positive expressions in the monocytes. In mild to severe UC, PD-1/PD-L1 expression was statistically significant ($P < 0.001$), and the expression level of PD-1/PD-L1 in mucosal tissues increased with disease activity, though there were still a few colon biopsies with negative PD-1/PD-L1 expression in the UC groups (Table 2).

## The expression of PD-1 on immune cells in the peripheral blood of UC patients, as analyzed by flow cytometry, was different from the healthy control group

The percentages of PD-1/T cells/CD4+T cells/CD8+T cells were statistically different between UC patients and healthy controls ($P < 0.05$), and the percentages of PD-1/T cells and PD-1/CD4+T cells in patients with moderate UC (PD-1/T 12.836.15% and PD-1/CD4

**Table 2 The percentage of PD-1/PD-L1 expression in monocyte and epithelium in healthy control, acute enteritis and UC.** The expression of PD-1/PD-L1was negative in healthy controls, while acute enteritis with very few PD-L1-positive expression in the monocyte. In mild to severe UC, PD-1/PD-L1 expression was statistically significant ($P < 0.001$), the expression level of PD-1/PD-L1 increased with disease activity.

| Group | PD-1 (monocyte) % | PD-L1 (epithelium) % | PD-L1 (monocyte) % |
|---|---|---|---|
| HC | 0 (0,0) | 0 (0,0) | 0 (0,0) |
| Acute enteritis | 0 (0,0) | 0 (0,0) | 0 (0,1) |
| Mild UC | 1.0 (0,4) | 1.0 (0,2) | 3 (3,6) |
| Moderate UC | 4.0 (1.25,7) | 3.5 (1.25,5) | 8.5 (7,11) |
| Severe UC | 9.0 (7,11) | 4.0 (2,5) | 9.0 (8,12) |
| Statistic | 81.919 | 79.185 | 106.182 |
| *P-value* | <0.001 | <0.001 | <0.001 |

Notes:

Comparisons of the median PD-1 monocyte staining percentage in pairs: HC *vs* Acute enteritis ($P = 1.000$); HC *vs* UC ($P=0.000$); Acute enteritis *vs* UC ($P = 0.000$); Mild UC *vs* Moderate UC ($P = 0.161$); Mild UC *vs* Severe UC ($P = 0.000$); Moderate UC *vs* Severe UC ($P = 0.207$).

Comparison of median percentage of PD-L1 epithelial cells staining by pairwise comparison: HC *vs* Acute enteritis ($P = 1.000$);HC *vs* UC ($P = 0.000$); Acute enteritis *vs* UC($P = 0.000$); Mild UC *vs* Moderate UC ($P = 0.114$); Mild UC *vs* Severe UC ($P = 0.026$); Moderate UC *vs* Severe UC ($P = 1.000$).

A pairwise comparison of the median percentage of PD-L1 stained monocytes: HC *vs* Acute enteritis ($P = 1.000$); HC *vs* UC($P = 0.000$); Acute enteritis *vs* UC ($P = 0.000$); Mild UC *vs* Moderate UC ($P = 0.017$); Mild UC *vs* Severe UC ($P = 0.002$); Moderate UC *vs* Severe UC ($P = 1.000$).

+T 19.67 9.95%) and patients with severe UC (PD-1/T 14.29 5.71% and PD-1/CD4+T 21.63 11.44%) were higher than in patients with mild UC (PD-1/T 8.17 2.80% and PD-1/CD4+T 12.44 4.73%; $P < 0.05$). There were no significant differences in PD-1/CD8+T cells between patients with mild to severe UC ($P > 0.05$), and the percentages of PD-1/CD8+T cells in patients with moderate to severe UC were higher than in the healthy controls ($P < 0.05$). These results are shown in Figs. 2A and 2B.

## Peripheral blood sPD-L1, but not sPD-1, was significant in UC patients

ELISA was used to measure sPD-1/sPD-L1 in peripheral blood to further analyze the function of sPD-1/sPD-L1. Although sPD-1 expression was increased in the severe UC group, there was no statistically significant difference in sPD-1 expression between the UC group and the control group ($P > 0.05$). However, the expression level of sPD-L1 was statistically different between the severe UC group and the control group (232.27 65.93 pg/mL), and sPD-L1 expression increased with UC disease severity ($P < 0.05$), from mild UC (256.38 80.23 pg/mL) and moderate UC (350.30 95.67 pg/mL) to severe UC (441.64 85.57 pg/mL), respectively (Fig. 3).

## A correlation analysis of sPD-L1 in peripheral blood found PD-L1 on monocytes or PD-L1 on epithelial cells in mucosal tissue

In order to further analyze the source of sPD-L1, we analyzed the correlation between sPD-L1 and PD-L1 on monocytes and PD-L1 on epithelial cells. The correlation coefficient between sPD-L1 and PD-L1 on monocytes was 0.606, and the correlation coefficient between sPD-L1 and PD-L1 on epithelial cells was 0.420 ($P < 0.001$; Table 3).

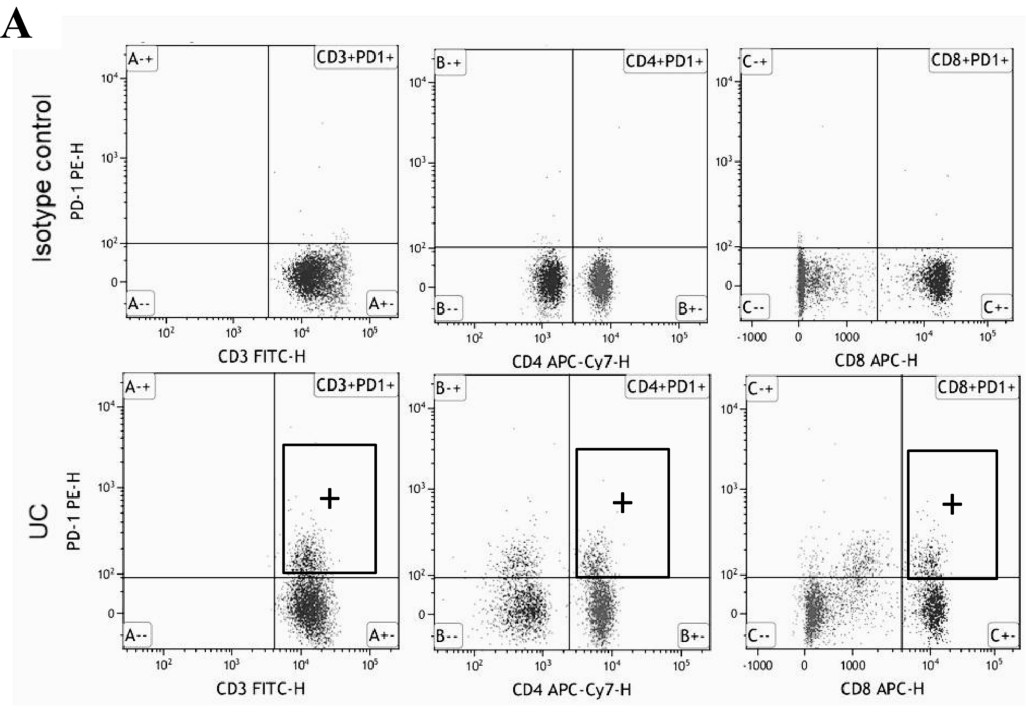

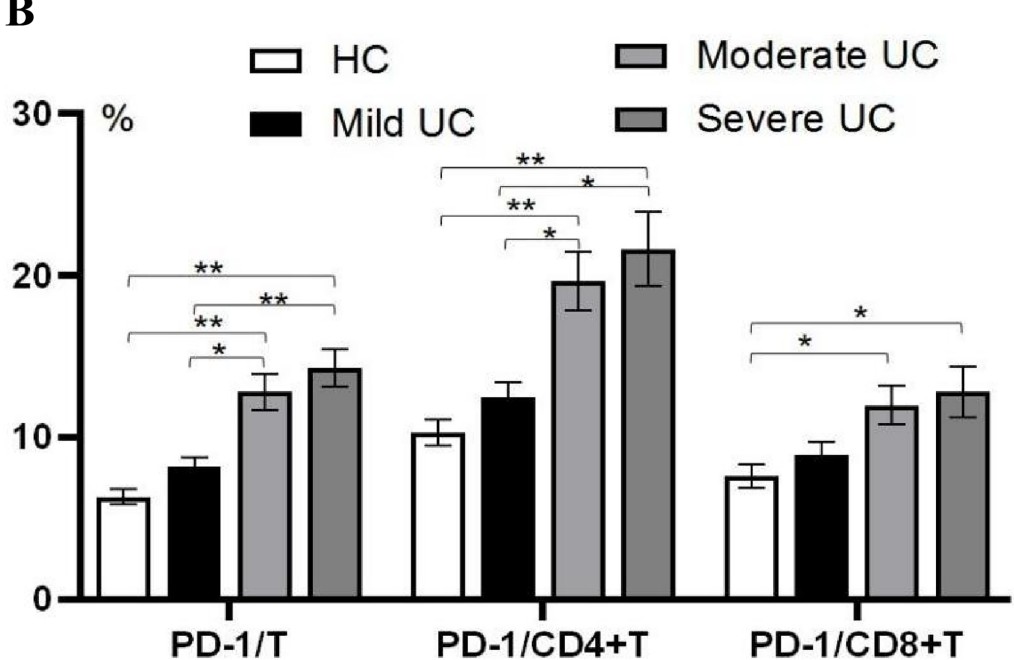

**Figure 2 The percentage of PD-1 on T cells,CD4+T cells and CD8+T cells in peripheral blood of UC and healthy control was analyzed by Flowcytometric.** (A) PD-1 on immune cells was analyzed by flow cytometry; (B) the expression of PD-1 on immune cells in UC patients with different disease degrees. CD3+PD1+:PD-1 positive expression on T lymphocytes; CD4+PD1+: PD-1 positive expression on CD4 +T lymphocytes; CD8+PD1+:PD-1 positive expression on CD8+T lymphocytes; PD-1/T:PD-1 positive expression on T lymphocytes; PD-1/CD4+T:PD-1 positive expression on CD4+T lymphocytes; PD-1/ CD8+T:PD-1 positive expression on CD8+T lymphocytes; $^{**}P < 0.001$; $^{*}P < 0.05$.

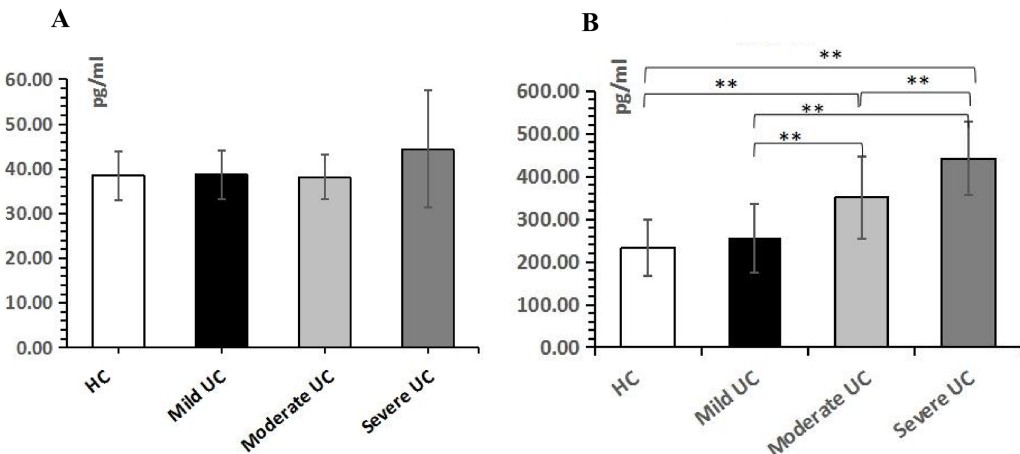

**Figure 3 Peripheral blood sPD-1/sPD-L1 in healthy control and ulcerative colitis patients.** (A) The expression level of sPD-1 in serum of UC patients. (B) The expression level of sPD-L1 in serum of UC patients. sPD-1: soluble Programmed cell death-1; sPD-L1: Programmed death-ligand 1; HC: healthy control; UC: ulcerative colitis; **$P < 0.001$.               

**Table 3 Correlation analysis of sPD-L1 with PD-L1 (monocyte) and PD-L1 (epithelium) in mucosal tissues.** The correlation coefficients between sPD-L1 and PD-L1 on monocytes, PD-L1 on epithelial cells is 0.606, 0.420, respectively ($P < 0.001$).

|   | PD-L1 (monocyte) | PD-L1 (epithelium) |
|---|---|---|
| r | 0.606 | 0.420 |
| P | <0.001 | <0.001 |

## A flow cytometry analysis of Th1/Th17 percentage in the peripheral blood of patients with ulcerative colitis

The expression level of Th1/Th17 was significantly different between UC patients and healthy controls, with Th1 cells increasing with the severity of the disease ($P < 0.05$), from mild UC (16.186.31%) and moderate UC (26.458.84%), to severe UC (30.8611.62%). Though Th17 expression levels were higher in UC patients than in healthy controls ($P < 0.05$), and Th17 expression increased with severity of illness, there was no statistically significant difference between mild UC (2.100.99%), moderate UC (2.881.46%), and severe UC (3.942.57%; $P > 0.05$; Fig. 4).

## Interferon levels in the peripheral blood of UC patients were significantly higher than in healthy controls, especially in the severe UC group

The content of interferon in the peripheral blood of mild UC patients (4.800.26 pg/mL), moderate UC patients (6.590.53 pg/mL) and severe UC patients (9.900.94 pg/mL) was higher than in healthy controls (2.940.12 pg/mL; $P < 0.001$), and interferon levels in the severe UC group were higher than in the mild UC group ($P < 0.005$; Fig. 5).

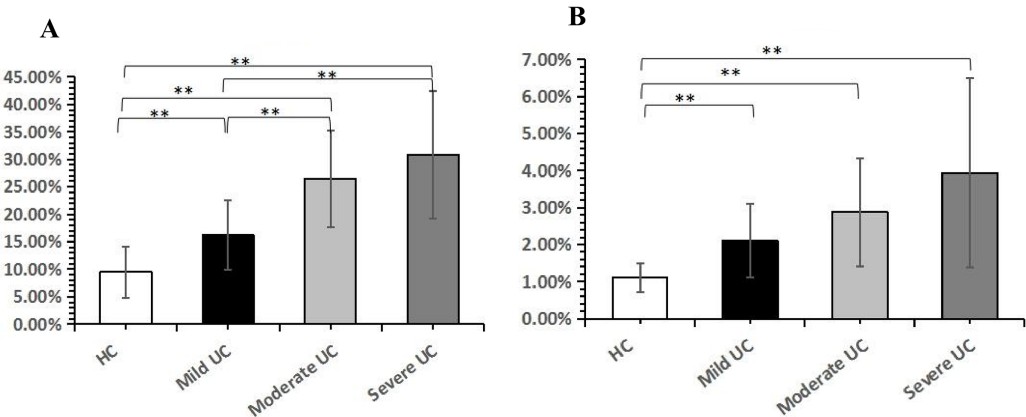

**Figure 4 The analysis of peripheral blood Th1, Th17 in ulcerative colitis patients.** (A) The expression level of Th1 in UC patients. (B) The expression level of Th17 in UC patients. Th1:T helper cell 1; Th17:T helper cell 17; $^{**}P < 0.001$. HC: healthy control; UC: ulcerative colitis.

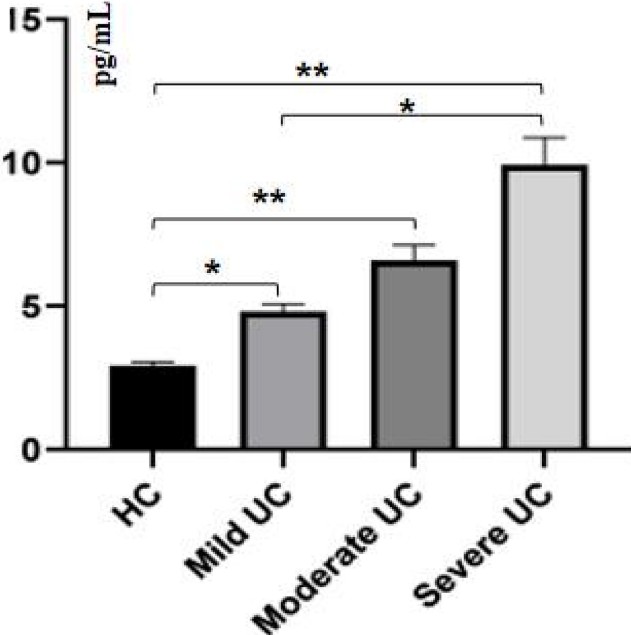

**Figure 5 Interferon was significantly increased in peripheral blood of the severe UC group.** $^{**}P < 0.001$; $^{*}P < 0.05$.

## A correlation analysis of PD-1/PD-L1 with peripheral blood Th1/Th17 showed a good correlation between Th1/Th17 and the expression of PD-1 on T cells

The results of the ELISA and flowcytometry analyses revealed positive correlations between PD-1/PD-L1 and Th1/Th17 (all $P$-values $< 0.005$). The correlation coefficients between Th1 and sPD-L1, PD-1/T, PD-1/CD4+T and PD-1/CD8+T were 0.427, 0.589, 0.486, and 0.329, respectively ($P < 0.001$). The correlation coefficients between Th17 and

**Table 4 Correlation analysis of PD-1/PD-L1 with peripheral blood Th1.** The correlation coefficients between Th1 and sPD-L1, PD-1/T, PD-1/CD4+T and PD-1/CD8+T is 0.427, 0.589, 0.486, 0.329, respectively ($P < 0.001$).

|   | sPD-L1 | PD-1/T | PD-1/CD4+T | PD-1/CD8+T |
|---|---|---|---|---|
| r | 0.427 | 0.589 | 0.486 | 0.329 |
| P | <0.001 | <0.001 | <0.001 | <0.001 |

**Table 5 Correlation analysis of PD-1/PD-L1 with peripheral blood Th17.** The correlation coefficients between Th17 and sPD-L1, PD-1/T, PD-1/CD4+T and PD-1/CD8+T is 0.323, 0.452, 0.320, 0.250, respectively ($P < 0.05$).

|   | sPD-L1 | PD-1/T | PD-1/CD4+T | PD-1/CD8+T |
|---|---|---|---|---|
| r | 0.323 | 0.452 | 0.320 | 0.250 |
| P | <0.001 | <0.001 | 0.001 | <0.05 |

sPD-L1, PD-1/T, PD-1/CD4+T and PD-1/CD8+T were 0.323, 0.452, 0.320, and 0.250, respectively ($P < 0.05$; Tables 4 and 5).

# DISCUSSION

This cross-sectional study of 80 UC patients found that PD-1 expression in monocytes is increased in UC patients compared to healthy controls and acute enteritis patients. PD-L1 expression in monocytes and epithelial cells was also increased, especially in the severe UC group. These results indicate that PD-1/PD-L1 expression tends to be up-regulated with an increase in UC disease severity, which differs from healthy controls and from an acute inflammation reaction.

In addition to being expressed primarily on activated T and B cells, PD-1/PD-L1 can also attenuate the activation signal of immune cells and mediate immune tolerance to autoantigens (*Bai et al., 2017*). In the clinical treatment of cancer, PD-1/PD-L1 has proven to be an important immunotherapeutic target, but 2–5% of tumor patients with anti-PD-1/PD-L1 may develop intestinal adverse reactions, and some patients experience structural changes to mucosal tissue, ulcers and other pathological changes similar to UC (*Han, Liu & Li, 2020; Dougan et al., 2021*). A previous study suggests that the interrupted PD-1/PD-L1 signal path contributes to the tolerance of intestinal mucosa to auto-antigens in mice, which could lead to severe autoimmune enteritis (*Chulkina, Beswick & Pinchuk, 2020*). Previous research also reflects the importance of PD-1/PD-L1 in maintaining intestinal mucosal health. In our study, we found up-regulation of PD-1/PD-L1 on inflammatory cells in the mucosal lamina propria and on epithelial cells. PD-1/PD-L1 was specifically expressed in UC mucosal tissues, with an increasing trend observed in PD-1/PD-L1 expression with the progression of inflammation, specifically the expression of PD-1/PD-L1 on monocytes in UC mucosal tissue, which is the result of the adaptation of mucosal tissue immune cells to chronic inflammation. However, not all UC tissue specimens in our study showed positive expression of PD-1/PD-L1. Some specimens (including samples

from the mild UC, moderate UC and severe UC groups) showed negative expression. There are two possible reasons for this. First, acute inflammation of UC mucosa was observed, and the number of neutrophils was much higher than that of monocytes. Second, the mucosal tissues of UC patients are in a sustained state of chronic inflammation, so the lymphocytes are no longer sensitive to the stimulation of cytokines and other signaling molecules.

To further assess the degree of immune dysfunction in UC patients, flow cytometry was used to detect Th1/Th17 in peripheral blood. Cellular immune dysfunction in UC patients was mainly observed in Th1/Th17, especially Th1, which increased with UC disease activity. Recent research has made clear that the dysfunction of lymphocyte subsets is a critical part of how UC develops immunologically (*Rovedatti et al., 2009*). CD4+ T cells are classified as helper or regulatory T cells, with a range of effector or regulatory functions. Aside from IFN-, other cytokines released by Th1 cells, such as TNF-α, and IL-17A, IL-17F, and IL-22, which is released by Th17 cells, play an important role in the immune response of UC patients (*Lee et al., 2021*).

Ulcerative Colitis is a chronic recurrent intestinal disease, and UC patients experience a state of sustained inflammation which activates the immune response of the body. Inflammatory factors secreted by immune cells further affect the expression of PD-1/PD-L1 on mucosal or peripheral blood immune cells. Some studies show that PD-1/PD-L1 mediates immune cell-macrophage interactions to control inflammation in the gut (*O'Malley et al., 2018*). There is evidence that B lymphocytes with high PD-L1 expression change from plasma cells into memory cells to affect the function of Th1/Th17 cells (*Khan et al., 2015*). *Aguirre et al. (2020)*, however, demonstrated that normal fibroblasts (MFs) can inhibit Th1/Th17 cell activity through the PD-1/PD-L1 pathway, while in Crohn's disease patients, increased matrix metalloproteinases can fracture PD-L1, contributing to Th cell dysregulation. This also indicates that the expression of PD-L1 in UC mucosal tissues is different from the expression in Crohn's disease. It is worth noting that PD-L1 can be cleaved by matrix metal enzymes, which is one of the important pathways for the production of sPD-L1. Studies have also found a positive correlation between PD-1 expression on Th cells and disease activity in active UC patients (*Long et al., 2021*). IFN-increases antigen presentation and promotes Th1 differentiation, leading to cellular immunity, as well as up-regulating PD-L1 in ovarian cancer cells, promoting tumour growth (*Abiko et al., 2015*). PD-1/PD-L1, which is influenced by cytokines, also regulates the function of immune cells. Another study suggests that IL-17 and TNF-α act individually rather than cooperatively to up-regulate PD-L1 expression in HCT116 cells by activating NF-κB and ERK1/2 (*Wang et al., 2017*). There is a soluble form of PD-1 found in the plasma of healthy individuals, and it is elevated in autoimmune diseases and chronic infections (*Khan, Arooj & Wang, 2021*). Excessive soluble PD-1 also blocks the PD-1/PD-L1 pathway, contributing to immunologic injury (*Zhao et al., 2018*; *Elhag et al., 2012*). Previous research has also shown that excessive amounts of soluble PD-1 contribute to the progression of arthritis *via* the Th1 and Th17 pathways (*Liu et al., 2015*). Differing from arthritis, sPD-1 levels in the peripheral blood of UC patients does not significantly differ from healthy controls, suggesting that sPD-1 may not play a role in UC. SPD-1 and sPD-

L1 have been detected in plasma, and elevated levels have been linked to advanced disease and poorer prognosis (*Khan et al., 2020*). However, the role of sPD-L1 in UC progression still needs to be explored. Our findings show that sPD-L1 levels in the peripheral blood of UC patients were significantly elevated and increased with UC disease severity. In a recent study, sPD-L1 was shown to inhibit T lymphocyte function, acting as a negative regulatory factor, indicating that sPD-L1 has a negative regulatory effect on immune cells in peripheral blood. SPD-L1 is also valuable in assessing disease severity in patients with UC. Previous research has investigated sPD-L1 as a biomarker of disease progression, prognosis, and response to checkpoint immunotherapy, and found that a high sPD-L1 level is associated with a worse clinical response (*Zhang et al., 2019*). Using peripheral blood sPD-L1 for UC prognosis needs further investigation, as our findings indicate sPD-L1 has the potential to evaluate the prognosis of patients with UC.

In combination with PD-L1, sPD-L1 is more of a general indicator of an inflammatory state, and the different forms of PD-L1 reinforce the dynamic crosstalk between the variety of cells implicated in the system (*Cheng et al., 2020*). The increased PD-1/PD-L1 in mucosal tissue and sPD-L1 in the peripheral blood of UC patients may function as protective feedback mechanisms made by immune cells in the state of inflammation. The exact factors that contribute to the dysregulated PD-1/PD-L1 balance in UC are not yet known, though this dysregulation increases the patient's susceptibility to autoimmune complications of UC. PD-1/PD-L1 is an important signaling molecule for future research on the pathogenesis of UC immunology and for identifying potential drug therapy targets.

## CONCLUSIONS

This study found that the expression level of PD-1/PD-L1 was correlated with UC disease activity, and two forms of PD-1 and PD-L1 may be used as potential diagnostic markers for UC and markers for assessing UC disease activity. PD-1/PD-L1 imbalance is a major characteristic of UC immune dysfunction. Future research should focus on the PD-1/PD-L1 signaling molecule and its connection to the pathogenesis of UC immunology and for identifying future drug therapy targets.

### Funding

A grant from the Health Commission of Shanxi Province financed this project (NO: 2020032). The funders had no role in study design, data collection and analysis, decision to publish, or preparation of the manuscript.

### Grant Disclosures

The following grant information was disclosed by the authors:
Health Commission of Shanxi Province: NO: 2020032.

### Competing Interests

The authors declare that they have no competing interests.

## Author Contributions

- Wei Shi conceived and designed the experiments, prepared figures and/or tables, and approved the final draft.
- Yu Zhang conceived and designed the experiments, performed the experiments, prepared figures and/or tables, and approved the final draft.
- Chonghua Hao conceived and designed the experiments, authored or reviewed drafts of the article, and approved the final draft.
- Xiaofeng Guo conceived and designed the experiments, authored or reviewed drafts of the article, and approved the final draft.
- Qin Yang performed the experiments, prepared figures and/or tables, authored or reviewed drafts of the article, and approved the final draft.
- Junfang Du performed the experiments, prepared figures and/or tables, and approved the final draft.
- Yabin Hou analyzed the data, prepared figures and/or tables, and approved the final draft.
- Gaigai Cao analyzed the data, prepared figures and/or tables, and approved the final draft.
- Jingru Li analyzed the data, prepared figures and/or tables, and approved the final draft.
- Haijiao Wang analyzed the data, prepared figures and/or tables, and approved the final draft.
- Wei Fang performed the experiments, prepared figures and/or tables, and approved the final draft.

## Human Ethics

The following information was supplied relating to ethical approvals (*i.e.*, approving body and any reference numbers):

The Shanxi Province People's Hospital granted Ethical approval to carry out the study within its facilities (Ethical Application Ref: 2021-18).

## Data Availability

The raw measurements are available in the Supplemental File.

## Supplemental Information

Supplemental information for this article can be found online at http://dx.doi.org/10.7717/peerj.15481#supplemental-information.

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
