# Peer review of "The significance of PD-1/PD-L1 imbalance in ulcerative colitis"

_PeerJ, doi:10.7717/peerj.15481_

## Round 0.1 · original submission · Major Revisions

According to the reviewer's opinion, this paper has several defects, including unreasonable experimental design and poor writing quality. The authors must carefully revise these comments.

Reviewer 1 ·

Basic reporting

PD-L1 is expressed on the surface of antigen-presenting cells such as DC cells, macrophages, and vascular endothelial cells when stimulated by IFN-γ. This article states that “both PD-L1 and PD-1 are expressed most frequently on activated CD4+ and CD8+ effector T cells, and their interaction inhibits activated CD4+ and CD8+T cell proliferation and mediates immune tolerance or cause a harmful effect on antitumor immunity, contributing to immune evasion.” However, the cited references did not specifically describe the expression of PD-L1 in immune cells. The authors need to double-check it.

Experimental design

1. As for the experimental design of this manuscript: There are 80 experimental groups and 30 healthy control groups. There is a significant difference between the experimental and control groups, and the number of subjects did not match statistically.

2. Regarding the flow cytometry analysis: I am wondering why the results of the experimental group (UC) and the healthy controls (HC) are available instead of the results of the acute enteritis group.

Validity of the findings

1. The author discovered that “PD-1 was merely expressed in monocyte located on mucosal lamina propria of UC patients” based on the experimental results. The authors did not specify what type of monocyte it was. In fact, immunohistochemical methods can be classified further. Why did the authors not continue their investigation?

2. In the discussion section, the authors stated that “we found that proinflammatory cytokines were up-regulated with the advancement of disease,” but in the experimental data section, it is stated that“Though Th17 expression levels were higher than controls (p<0.05), and Th17 increased with severity of illness, there was no statistically significant difference among different disease degrees (p>0.05) (Fig. 5)”.Data that are not statistically significant cannot be written as a positive result in the discussion section.

3. Although the manuscript stated that “although there were still a few colon biopsies with negative expression of PD-1/PD-L1 in UC group” in the immunohistochemistry results, the potential causes were not discussed in the discussion section.

Additional comments

There have been many studies on the pathogenesis and clinical applications of ulcerative colitis, but there have been very few studies that can apply PD-1 and PD-L1 to ulcerative colitis. Therefore I think the conceptualized idea of this manuscript is novel.

PD-1 and PD-L1 are thoroughly examined from two distinct effector sites, peripheral blood and local pathological tissue, as well as membrane-bound and soluble PD-1 and PD-L1 (the two different forms of action). Flow cytometry, ELISA, and immunohistochemistry are three different detection methods used to determine the relationship between PD-1/PD-L1 and UC. As a result, the experimental design in this article is rigorous.

However, the article has some flaws, which have been listed above. I would appreciate it if the authors could make changes in accordance with the above points so that this manuscript could be more in line with the publishing requirements.

Reviewer 2 ·

Basic reporting

In an effort to evaluate immunotherapy to treat ulcerative colitis (UC), Shi et al investigated the expression and function of Programmed Cell Death-1 (PD-1) and Programmed Cell Death ligand-1 (PD-L1) in both mucosal tissues and peripheral blood from UC patient samples. The authors concluded that PD-1 and PD-L1 are localized differentially in UC samples, and that the expression of PD-1 and PD-L1 is correlated with severity of the disease state. The study also showed moderate correlation between PD-1 and PD-L1 expression in functional T cells in the peripheral blood with the disease state. However, only soluble PD-L1, not soluble PD-1 was considerate significantly correlated with UC disease state.

Major improvement is needed on the writing of this article before it can be considered at publication level. The structure meets the journal standards, and figures are well represented, and raw data and related paperwork on human tissue research is provided clearly. however, the figures need to be labeled better and described.

Experimental design

The authors analyzed patient samples with various UC disease states and acquired across a good span of patients in age and gender. HE staining, flow cytometry and ELISA assay is employed to examine the expression of PD-1 and PD-L1 in appropriate environments and in correlation with disease severity of patients. The knowledge will provide important baseline information towards a potential therapeutic resolution of UC treatment.

Validity of the findings

The study was well controlled for and the conclusions are supported by proper and sound statistical analyses.

Additional comments

Major comments:
1. The writing of the article needs major revisions:
a. Grammatical errors appear sporadically and needs tidying up.
b. The results sections should be titled with conclusions but not methods; they also need context for rationale for experimental design. Importantly, they cannot be the same as language used in figure legends.
c. Figure legends need to be described more in detail.
2. Figures 2 and 3 appear to be on very closely related subjects and is mentioned together in the main text. In such case it is recommended that these two figures are combined as different panels of the same figure.
3. The authors suggested, and backed up by literature, that ulcerative colitis disease severity and PD-1/PD-L1 expression is correlated with Th1 and Th17 cells. To confirm the functional relevance of and reveal the mechanisms behind such correlations, it is highly recommended to also examine the downstream cytokine levels in corresponding samples.

Minor comments:
1. Error bar is need for Figure 3.
2. Certain contents in the discussion section appears tangential and can be shortened for better focus around findings from this study.
3. It is recommended that the authors propose further hypotheses to address the discrepancies between sPD-1 and sPD-L1 regarding their correlation with disease severity in the peripheral blood.

Reviewer 3 ·

Basic reporting

The manuscript by She et al examines the expression PD-1 and PD-L1 in individuals with ulcerative colitis. According to their study, the expression of these two molecule were higher among this population comparing to healthy people and the population with acute enteritis. Despite the interesting topic this study has some drawback for publication:
First of all the English language of the study needs extensive revision. In some section of the study, specially the result section, some sentences lacked verb! and it seems that the author only pasted the figure legends in the results. Besides, some of the sentences were started with conjunctions such as "and". Moreover, the paragraphs were poorly structured in the some sections such as result and discussion.
Secondly, it should be noted that some studies previously reported the expression pattern of PD-1 and PD-L1 in individuals with ulcerative colitis. The results of such studies should be mentioned in the introduction section. This is of high importance to do so, since the introduction section is the main place of addressing gap of knowledge or any hypothesis about one topic and simultaneously introduce the possible solutions for those gaps. However, I could not find any such explanation in the introduction section. Hence, I suggest the authors add the current literature to the introduction section and highlight the gaps of the topic specifically.
Also check all the abbreviation through the manuscript. Some of them such as sPD-1 was not defined in the text and in some other instances the whole phrase of word mentioned once and still the author introduced an abbreviation for that. What does CD patients stand for in line 193?
In the discussion section on, I was wondering why the authors mainly cited references concerning about the expression PD-1 and PD-L1 in different cancer while there are many studies check the expression PD-1 and PD-L1 in various autoimmune diseases.
However, I should appreciate the authors for sharing the data which provide the reviewers and editors to validate the study findings.

Experimental design

At the end of the introduction section the authors stated that the purpose of this study to show the role of PD-1 and PD-L1 in pathogenesis of ulcerative colitis. However, for such strong statement more experiments were needed. In fact in the discussion section the author clearly claimed that " We do not yet know what factors contribute to the dysregulated PD-1/PD-L1balance and can increase susceptibility to autoimmune complications of UC" which some how implied that the authors, themselves, believed the their study could not show the role of PD-1/PD-L1 in pathogenesis of UC.
Moreover, in the method section there were some fundamental details missed:
1. It was unclear how the authors recruited patients in different severity of the UC while none received any immunosuppression before? Were the patients’ disease newly diagnosed at the time of the study?
2. Is there any matching for the case and controls? Apparently and based on the details provided by the authors, the place where the study was conducted, it is possible to find match controls but I could not find any details regarding matching.
3.How did the authors define the acute enteritis?
4. Under what indication do the authors obtain tissue samples from the healthy subjects? The same question can be true for the subjects with the acute enteritis.
5.I was wondering how the author came with the 80 patients sample size? Was there any sample size calculation? Since the sample size of each group was so round, I assume that the authors assign the sample size of each group.
6. In the statistical section, the authors mentioned any details about neither the correlation methods nor the post hoc analysis?
Finally I suggest the author to revise the method section based on STROBE or REMARK guidelines

Validity of the findings

1. The overall structure of the result section need to revised. in paragraph one of the result section, the authors use a similar style of reporting patients characteristics for all study groups.
2. I suggest the authors add more data (other than sex and age) and prevent similar writing structure. No need to mention everything for all study groups. Just focus on key features and address the rest in table 1.
3. Some sections need to fully rewritten since apparently in some sections of the result, the author just paste the figure legends. The whole structure of the tables were fine, however, what does statistics refer to in table 2.
4. What are numbers in the parentheses table 2? apparently they indicate range of each variable. if so, please add such detail to the caption of the table.
5. If possible please perform the correlation analysis between the ELISA and IHC findings for sPD-L1 and epithelial PD-L1.
6. Conclusions are poorly described. In fact with the respect to sample type, only the serum level of PD-L1 can be considered as the biomarker for UC.

Additional comments

1. What does experimental group refer to in line 148?
2. If possible please add a paragraph to discussion about the possible application of the PD-1 and PD-L1 in management of UC.

---

## Round 0.2 · Minor Revisions

Reviewer 2 proposed several minor comments. Thank you.

Reviewer 1 ·

Basic reporting

.

Experimental design

.

Validity of the findings

.

Additional comments

My comments have been addressed by the authors. I think the manuscript has been significantly improved. Therefore, I would like to suggest the acceptance of this manuscript.

Reviewer 2 ·

Basic reporting

The authors have addressed the vast majority of reviewer's comments. The language is significantly improved and proper conclusions and discussions were added which strengthened the findings. I find it now meets the standard for publication.

Experimental design

The authors have addressed the vast majority of reviewer's comments.

Validity of the findings

The authors have addressed the vast majority of reviewer's comments. All data supported the findings from this study

Additional comments

Figure legends were not described in details sufficiently in that the breakout panels (A,B...) were not referred to when describing the figures. The paragraph titles in the results section, while truthfully reflects the findings, may appear too much in details, and could be trimmed back to only present a brief one-sentence major finding.

Reviewer 3 ·

Basic reporting

In the revised version of the manuscript, the authors have addressed many of my concerns. The English language of the study has been improved substantially. However, they did not include the studies assessing the expression of PD-1/PD-L1 in UC patients in the introduction, due to scarcity of these studies!

Experimental design

I have no further comments in this section.

Validity of the findings

I have no further comments in this section.

---

## Round 0.3 · accepted · Accept

After revision, the current version can be accepted.